# Quantifying population contact patterns in the United States during the COVID-19 pandemic

Dennis M. Feehan ⬤ [1✉] & Ayesha S. Mahmud ⬤ [1✉]

SARS-CoV-2 is transmitted primarily through close, person-to-person interactions. Physical distancing policies can control the spread of SARS-CoV-2 by reducing the amount of these interactions in a population. Here, we report results from four waves of contact surveys designed to quantify the impact of these policies during the COVID-19 pandemic in the United States. We surveyed 9,743 respondents between March 22 and September 26, 2020. We find that interpersonal contact has been dramatically reduced in the US, with an 82% (95%CI: 80%–83%) reduction in the average number of daily contacts observed during the first wave compared to pre-pandemic levels. However, we find increases in contact rates over the subsequent waves. We also find that certain demographic groups, including people under 45 and males, have significantly higher contact rates than the rest of the population. Tracking these changes can provide rapid assessments of the impact of physical distancing policies and help to identify at-risk populations.

[1] Department of Demography, University of California, Berkeley, Berkeley CA, USA. ✉email: feehan@berkeley.edu; mahmuda@berkeley.edu

The dynamics of COVID-19 in a population are fundamentally dependent on rates of interpersonal interaction and on patterns of who interacts with whom. With the sharp increase in COVID-19 cases globally, many countries adopted physical distancing practices at an unprecedented scale in an effort to reduce transmission. On 16 March 2020, seven counties in the San Francisco Bay Area ordered residents to shelter in place in response to evidence of community transmission of COVID-19. Over the subsequent days and weeks, other US cities and states followed suit. At the start of April 2020, the majority of people living in the US were under orders to dramatically restrict their daily activities. By the end of April, however, some localities began easing restrictions, and there is presently considerable heterogeneity in physical distancing policies across US states, counties, and cities[1].

Strong physical distancing measures are effective in controlling the spread of the virus only if they are able to reduce the amount of close interpersonal contact in a population. To quantify how much interpersonal contact is changing as the pandemic evolves in the US, we developed the Berkeley Interpersonal Contact Survey (BICS). The BICS study collects information about the total number of contacts people have, as well as detailed information about who people are interacting with. This detailed information is particularly important for informing epidemiological models and for identifying populations at greatest risk to COVID-19. Age-structured contact rates are especially relevant for COVID-19 because of age-related variation in clinical outcomes, and possibly susceptibility and transmissibility[2].

Here, we describe changes in contact rates and patterns over the course of the pandemic, and identify important correlates of close interpersonal contact in the US. We also evaluate the effectiveness of physical distancing policies by estimating the impact of reduced contact rates on the reproduction number, $R_0$—the average number of secondary infections arising from a single infection in a fully susceptible population.

## Results

**Data collection.** Data collection took place in four waves: between 22 March and 8 April 2020 (pilot study, Wave 0); between 10 April and 4 May 2020 (Wave 1); between 17 and 23 June 2020 (Wave 2); and between 11 and 26 September (Wave 3). We surveyed a total of 9743 respondents in the US (Wave 0 $n = 1437$, Wave 1 $n = 2627$, Wave 2 $n = 2431$, Wave 3 $n = 3248$). Survey respondents were asked to report the number of people they had contact with on the day before the interview. Respondents reported a total of 49,321 contacts and provided detailed reports about 29,880 contacts. We oversampled respondents in certain cities; analyses here are weighted to account for sample composition ("Methods").

**Interpersonal contact in the United States.** Since physical distancing policies are intended primarily to reduce non-household contacts, we investigate both the total number of reported contacts and the number of reported non-household contacts. Fig. 1a, b show histograms of the number of contacts (Fig. 1a) and non-household contacts (Fig. 1b) reported by respondents in each wave. Respondents reported a median of two contacts (0 non-household) in Wave 0, a median of three contacts (1 non-household) in Wave 1, a median of three contacts (1 non-household) in Wave 2, and a median of four contacts (2 non-household) in Wave 3. Qualitatively, the pattern of contacts is similar in each wave, but with increasingly higher levels of contact in Waves 1, 2, and 3, when compared to Wave 0. We confirm this increase in contact levels over time with a model-based analysis below.

For up to three contacts, respondents were asked to report detailed information, including the contact's age, sex, relationship to the respondent, and the location of the contact event. Using this information, we estimated the composition of respondents' contacts by relationship and by location (see "Methods"). Figure 1 shows the estimated average number of non-household contacts each person reported to have taken place by contact's relationship (Fig. 1c) and location (Fig. 1d). These are contacts respondents

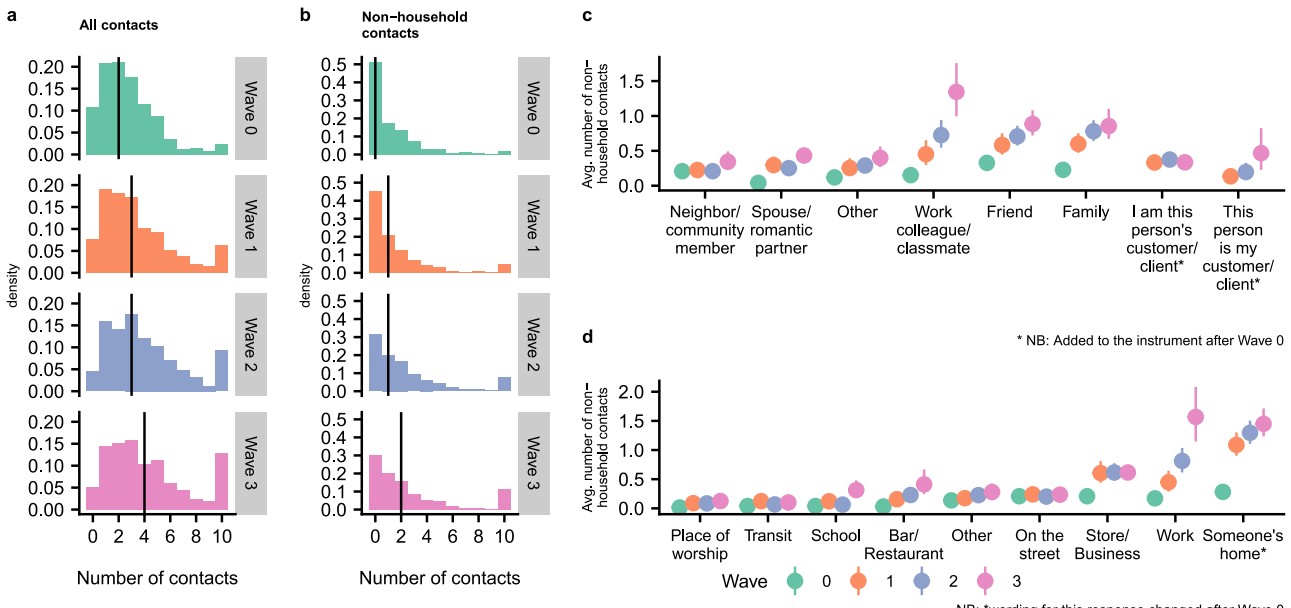

**Fig. 1 Reported interpersonal contact across four survey waves. a, b** Histograms of reported number of contacts (**a**) and non-household contacts (**b**) among respondents for each wave. Reported contacts are topcoded at 10 in these plots. The vertical lines show the median number of contacts. **c, d** Estimated average number of non-household contacts each person reported to have taken place by contact's relationship (**c**) and location (**d**) based on $n = 29,880$ reports about detailed contacts ("Methods"). Uncertainty estimates are 95% intervals derived from the bootstrap. Each point shows estimated average numbers of non-household contacts in each category, per person. For example, Panel **c** shows that the average respondent reported almost 0.8 non-household contacts with family members in Wave 2. Avg. average.

reported with people who do not live in their household. It is therefore possible that some of these "home" contacts took place in the respondent's household; this would happen if, for example, neighbors came over to visit. They could also have taken place in someone else's household as would happen if, for example, the respondent had visited a friend at the friend's house. Across Waves 0 to 3, the average number of interactions with family, friends, and work colleagues increases, and in Waves 1 to 3, these three relationships are responsible for most non-household interpersonal contact. In Wave 0, with contact levels uniformly very low, no single relationship stands out as explaining most non-household interaction. Across Waves 0 to 2, the most common location of reported contacts was someone's home; by Wave 3, work and home had similar levels of reported contacts. Across Waves 0 to 3 we find increases in the number of work contacts and home contacts, and between Waves 0 and 1 we see increases in contacts at stores and businesses.

Previous studies have found that during non-pandemic periods the average number of contacts is related to characteristics of people—e.g., age and household size—and to structural factors like day of the week—weekday versus weekend[3]. To investigate correlates of contacts in the US during the emergence of COVID-19, we fit negative binomial regression models to data for all contacts and to data for non-household contacts (see "Methods"). Figure 2 summarizes inferences from the model for non-household contacts by showing conditional effects plots for different covariate values (see Supplementary Table 4 for the posterior mean estimates and 95% credible intervals for all coefficients from the two models). These conditional effects plots show the expected number of non-household contacts and the 95% posterior credible interval for different covariate values; covariate values not being manipulated in each panel are fixed at the values for a white female aged 35–44 from the national sample who lives in a two-person household during a weekday in wave 3. For example, Fig. 2a compares the predicted number of

non-household contacts on a weekday and on a weekend for a white female aged 35–44 from the national sample who lives in a two-person household during wave 3 (Supplementary Fig. 2 shows analogous results from a model fit to all contacts).

Several interesting findings emerge from Fig. 2. The model estimates confirm that the average level of non-household contact increased with each wave, but the pace of this increase varied by city: for example, model estimates suggest that contact rates in the Bay Area and Phoenix steadily climbed from Wave 0 to Wave 3; in contrast, other cities—including Atlanta, Boston, New York, and Philadelphia—saw uneven increases in contact levels from Wave 0 to Wave 3. Patterns of contact rates by race/ethnicity also vary over time: in Wave 1, Black and Hispanic respondents reported highest contact rates, but by Wave 3, Whites reported the highest contact rates. Respondents under age 45, especially males, report higher contact rates than older respondents. There is little evidence for differences in numbers of non-household contact by day of the week or household size. The Supporting Information contains additional analyses of contact patterns.

**Implications for COVID-19 transmission.** To estimate relative changes in transmission over the course of the pandemic, we estimated the impact of changing contact rates on the reproduction number. According to the social contact hypothesis, for respiratory pathogens such as SARS-COV-2, relative changes in $R_0$, can be estimated by comparing the dominant eigenvalues of age-structured contact matrices[4,5]. Note that our estimated reproduction number for each time point indicates the transmission potential for the pathogen in a fully susceptible population; we cannot directly estimate the time-varying effective reproductive number—the average number of secondary infections per case at each time point in the epidemic—without additional information on the fraction of population that is susceptible. Thus, our estimated $R_0$ value at each time point represents the theoretical $R_0$ for an outbreak in a fully susceptible

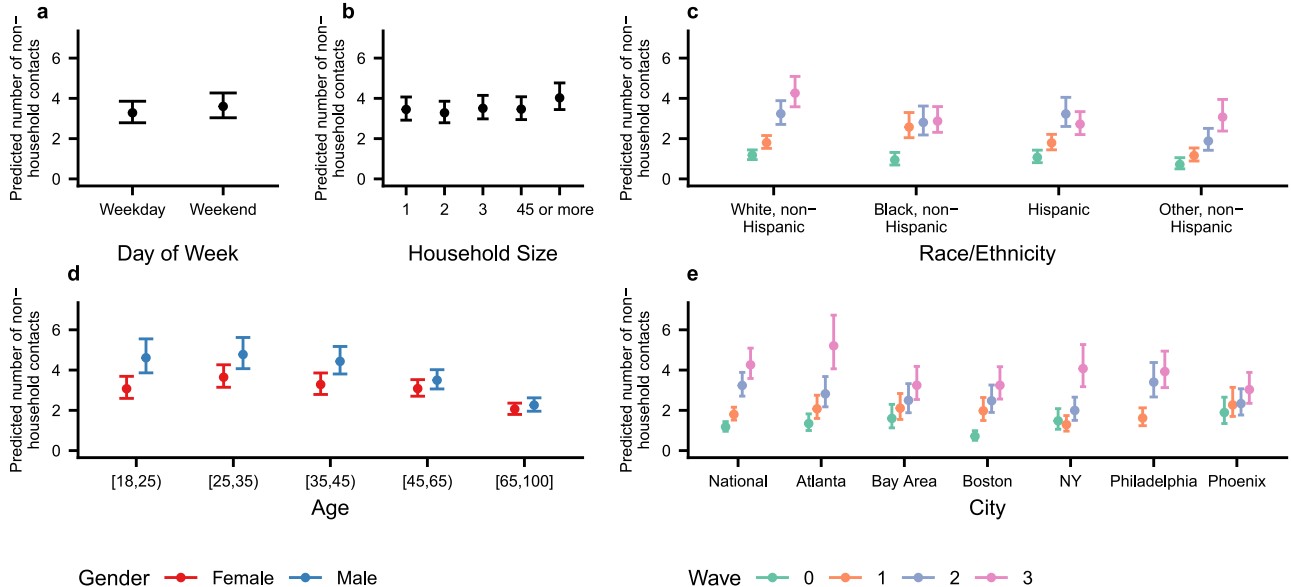

**Fig. 2 Conditional effect plots showing the predicted mean number of non-household contacts and 95% posterior credible intervals for several covariates.** Predicted mean number of non-household contacts is shown for (**a**) day of the week; (**b**) household size; (**c**) race/ethnicity; (**d**) age/sex group; and (**e**) geography. Predictions come from a negative binomial model fit to reported numbers of non-household contacts made by n = 9743 survey respondents. Colors are used in panels **c–e** to show estimated interactions. Covariate values not being manipulated in each panel are set to values for a white female aged 35–44 from the national sample who lives in a two-person household during a weekday in wave 3 ("Methods"). Uncertainty bars show 95% posterior credible intervals. Supplementary Fig. 2 shows the same predictions for an analogous model fit to all contacts.

population subject to the observed age-structured contact matrix at that time point.

We calculated age-structured contact matrices, adjusting for the age distribution of survey respondents and the reciprocal nature of contacts, for each wave of the BICS study (see "Methods"). We compare these with baseline data on pre-pandemic contact patterns in the US to understand the impact of physical distancing policies on contact rates and the implications for the transmission of SARS-CoV-2. There are surprisingly few existing estimates for the rate of contact in the US before the COVID-19 pandemic[6–8]; here, we compare our estimates to contact patterns estimated from a probability sample of US Facebook users in 2015 (ref. [9]) (see Supplementary Fig. 7 for a comparison of available pre-pandemic estimates of contact patterns).

We find large declines in daily interpersonal interaction compared to business as usual, with the largest decline in Wave 0 (82%) followed by Wave 1 (74 %), Wave 2 (68%), and Wave 3 (60%). Figure 3 shows the estimated age-structured contact matrix and the reduction in interpersonal contact in each age category for the four BICS waves compared to the 2015 study. We find considerable declines across all age groups, particularly in Wave 0, with largest absolute decline in the 25–35 age group. However, even at these low absolute levels of interpersonal contact, we continue to find distinctive patterns of assortative mixing by age found in previous contact studies.

We estimated the relative reduction in $R_0$, assuming (1) that contact patterns in the population before physical distancing became widespread were equivalent to the 2015 study[9] and (2) that disease-specific parameters remained unchanged over the course of the survey period (see "Methods"). We find 73% (95% CI: 72–75%), 57% (95% CI: 53–61%), 48% (95% CI: 43–53%), and 36% (95% CI: 29–42%) declines in the implied $R_0$ in Waves 0, 1, 2, and 3 respectively, relative to the pre-pandemic period. The contact patterns observed in our survey suggest a substantial reduction in $R_0$ under physical distancing, particularly during the Wave 0 study period. Figure 4 shows the $R_0$ estimates for the four survey waves, assuming an average $R_0$ value of 2.5 in the absence of physical distancing. The dramatic reduction in contact rates observed in Wave 0 was sufficient in reducing $R_0$ to 0.66 (95% CI: 0.38–0.96) in Wave 0. However, with the easing of physical distancing and increase in overall contact rates, $R_0$ increased to 1.06 (95% CI: 0.61–1.53) by Wave 1, 1.29 (95% CI: 0.74–1.86) by Wave 2, and 1.59 (95% CI: 0.91–2.30) by Wave 3. We repeat the analysis using contact patterns from UK participants in the POLYMOD study[3], which has been the gold-standard for modeling age-specific contact patterns in many settings, as the pre-pandemic baseline; our results are qualitatively similar (Fig. 4).

While physical distancing reduces the risk of transmission by reducing contact rates in the population, there is evidence to suggest that the adoption of other non-pharmaceutical interventions, such as the usage of face coverings or masks, can further reduce transmission. To account for this, we repeated the analysis by restricting contacts to only those where no mask usage was reported (Fig. 4). Accounting for mask usage reduces the relative increase in the implied $R_0$. The two scenarios modeled here represent the extreme ends of the possible spectrum of protection conferred by mask usage, i.e., from no efficacy to perfect efficacy in reducing transmission; actual $R_0$ is likely to fall within these two bounds.

## Discussion

We find large reductions in the number of contacts reported in our survey compared to business as usual, suggesting that the physical distancing measures adopted in the US in March and April had their intended impact. Compared to the contact survey conducted in 2015 (ref. [9]), our estimates suggest that in Wave 0 there was about 82% (95% CI: 80–83%) reduction in the daily average number of contacts per person. This finding is similar to the declines in contact rates, relative to pre-pandemic levels, recently observed elsewhere; 86% decline in Wuhan, China, 88% decline in Shanghai, China[10], 74% decline in the United Kingdom[5], 82% decline in Luxembourg[11], 85% in Italy and between 73 and 75% decline in Italy, Belgium, France, and the Netherlands[12].

As time elapsed, physical distancing policies were relaxed and then, in some jurisdictions, reimposed. We find that over this time period the rate of close interpersonal contacts in the US gradually increased from an unprecedented low level in March, pushing the estimated $R_0$ values above 1 by June. In addition to an overall increase in the average number of reported contacts, we also find an increase in the number of contacts at work, as well as at stores and businesses; this has implications for SARS-CoV-2 transmission as the economy reopens.

Our analysis here has several important limitations. In this study, we used a quota sample from an online panel rather than a probability sample. Previous contact studies have also used various alternatives to probability samples[13–17]. Online panels allow data to be collected rapidly and frequently, whereas the time and cost required to design and implement a probability sample are prohibitive. Further, obtaining a probability sample during a pandemic is complicated by the logistical challenges arising from the need to protect interviewers and respondents. However, future work based on a national probability sample would be a valuable complement to our study.

There may be some recall bias in our survey estimates, as respondents were asked to report on contacts from the previous day. There may also be social desirability bias arising from awareness of social distancing policies. Our surveys were only conducted in English, meaning that we are not able to reach people who only speak other languages. We do not survey children, and are unable to capture contacts within age groups below the age of 18. Finally, our estimates of relative changes in $R_0$ do not take into account possible age-specific differences in susceptibility or infectiousness, or possible changes in infection transmissibility due to other factors.

The BICS study is ongoing, and will continue to collect data for the next several months, with the goal of measuring changes in contact patterns as interventions change and schools and workplaces reopen. The data from the BICS study provide a unique opportunity to understand how interpersonal contact patterns are changing in the US over the course of the pandemic, and the epidemiological implications for COVID-19 and other respiratory pathogens. Future work will focus on applying these estimates to parameterize age-structured mathematical models of SARS-CoV-2 transmission and to monitor and evaluate the effectiveness of physical distancing policies over time.

## Methods

**Survey methodology**. We designed and fielded a survey to measure interpersonal interaction in the United States. Following the POLYMOD project[3] and subsequent studies[5,10,18], survey respondents were asked to report the number of people they had conversational contact with on the day before the interview; in Waves 1 to 3, we also asked about physical contact. Respondents were asked to provide detailed information about up to three of their reported contacts; this detailed information included who those contacts were, how long those contacts lasted, and where they took place. In Wave 0, respondents were asked to report all contacts, and to then report how many of their contacts were not household members. Starting with Wave 1, respondents were asked to provide a household roster, and then report only contacts outside of the household.

The survey instrument was created in Qualtrics and respondents were recruited using Lucid, an online panel provider. In each wave, we obtained two samples: first,

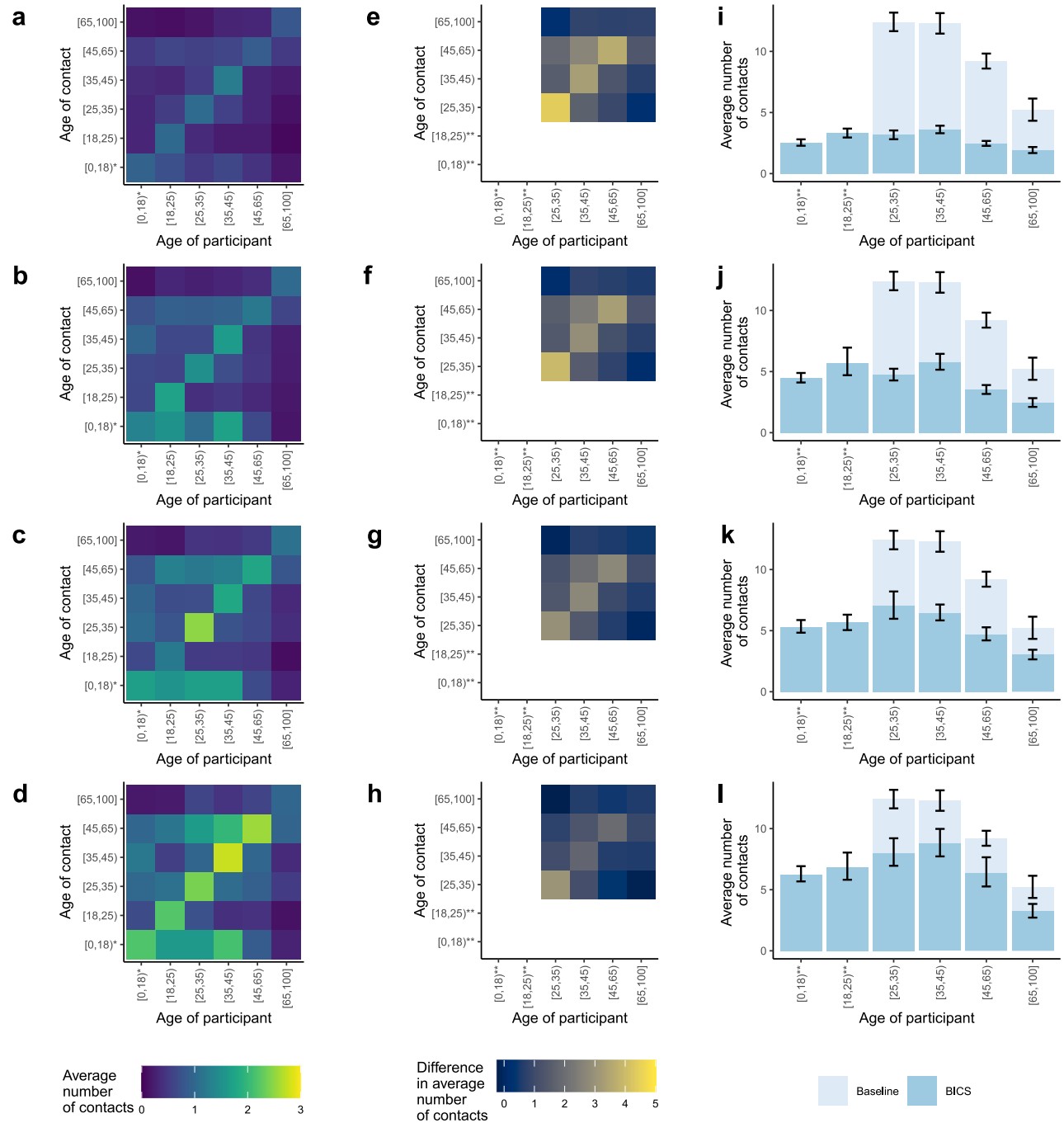

\* NB: Within−group mixing for the [0,18) group was estimated using POLYMOD UK data
\*\* NB: Youngest age−groups had different widths in the baseline survey

**Fig. 3 Comparison of age-structured contact matrices with baseline. a–d** Age-structured contact matrices from the four BICS waves after adjusting for the age distribution of survey respondents and the reciprocal nature of contacts; lighter colors indicate higher number of average daily contacts. **e–h** Difference in the average number of contacts between the 2015 study and the four BICS waves; lighter colors indicate a larger absolute difference between the 2015 study and the BICS data. **i–l** Average number of reported contacts for each respondent age group for the BICS data (darker color) compared to the 2015 study (lighter color), along with 95% confidence intervals derived from the bootstrap. The BICS estimates are based on $n = 3163$ in Wave 0, $n = 7473$ in Wave 1, $n = 7842$ in Wave 2, and $n = 11,402$ in Wave 3 reported contacts; The 2015 study estimates are based on $n = 5944$ reported contacts. Top row shows BICS Wave 0; second row shows BICS Wave 1; third row shows BICS Wave 2; and bottom row shows BICS Wave 3.

a quota sample that is intended to be representative of the United States; and, second, several smaller quota samples from specific cities: New York, the San Francisco Bay Area, Atlanta, Phoenix, and Boston. In Wave 1, Philadelphia was added.

All survey respondents provided informed consent and the project was approved by the UC Berkeley IRB (Protocol 2020-03-13128).

**Weighting**

*Respondent-level weights.* We adopt a model-based approach to inference, which is appropriate for our quota sample[19]. Except where noted, we pool results from the national and city samples together in this analysis. We use calibration to produce pseudo-probabilities of inclusion, and use these pseudo-probabilities of inclusion as the basis for weights used to make population-level inferences[20,21]. We calibrate

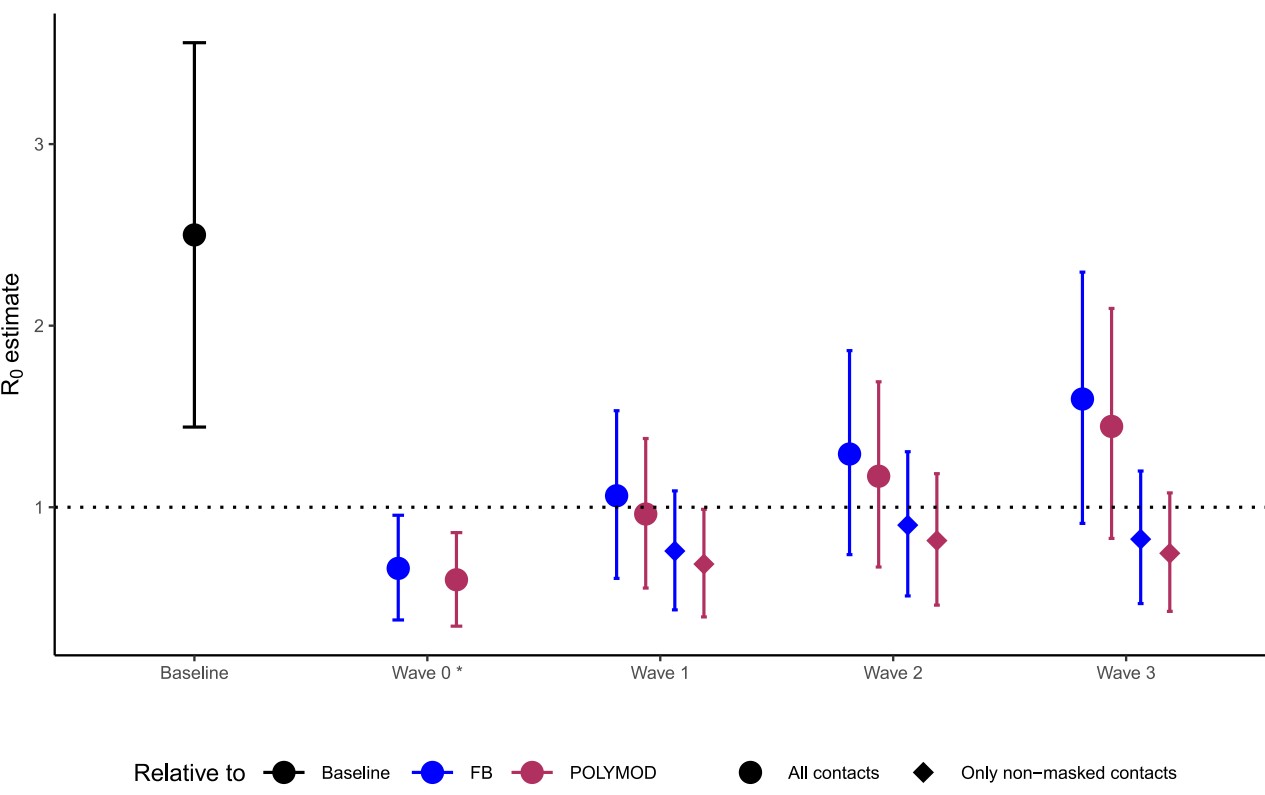

* NB: Face mask usage data was not collected in Wave 0

**Fig. 4 Implied $R_0$ estimates for each wave.** The implied $R_0$ from the BICS contact matrices for each wave relative to two baseline contact matrices from the 2015 study and the UK POLYMOD study, and assuming a baseline $R_0$ value drawn from a normal distribution with mean 2.5 and standard deviation of 0.54. Circles indicate $R_0$ estimates calculated from age-structured contact matrices for all reported contacts ($n = 3163$ in Wave 0, $n = 7473$ in Wave 1, $n = 7842$ in Wave 2 and $n = 11,402$ in Wave 3); diamonds indicate $R_0$ estimates calculated from age-structured contact matrices for contacts where no mask usage was reported ($n = 5777$ in Wave 1, $n = 5818$ in Wave 2 and $n = 7583$ in Wave 3). Ninety-five percent confidence intervals were derived from the bootstrap. FB: 2015 Facebook survey.

based on: age categories (18–23, 24–29, 30–39, 40–49, 50–59, 60–69, 70+); sex; age by sex interactions; education (non-high school graduate, high school graduate, some college, college graduate); race (white, Black, other); Hispanicity; household size category (1, 2, 3, 4, 5, or more); and whether the respondent's county is rural/suburban/urban. Figure 5 shows the distribution of respondents before and after calibration weighting. All population values except for rural/suburban/urban are taken from a 1-year extract of the 2018 American Community Survey provided by IPUMS[22]. We ascertain whether each respondent lives in an urban, suburban, or rural area by mapping the respondent's zip code to county, and then using the county-level urban/suburban/rural codes from the CDC. In order to map zip code to county, we use the crosswalk developed by Sood[23]. We perform the calibration using the R packages autumn (https://github.com/aaronrudkin/autumn) and leaf-peepr (https://rdrr.io/github/rossellhayes/leafpeepr/).

*Contact-level weights.* In Wave 0, the pilot study, respondents were asked for their total number of contacts and for the number of contacts who were not household members. Then, respondents were asked to provide detailed information for three of these contacts; this detailed information included contact age, sex, relationship to respondent, and contact location. If respondents reported more than three total contacts, they were asked to report in detail about the first three contacts who came to mind. Starting with Wave 1, respondents were asked to report about the age and sex of all of their household members, and then to report the number of contacts they had with non-household members. Respondents were then asked to report detailed information for the first three non-household member contacts who came to mind.

In all waves, some respondents reported more than three total contacts, but only provided detailed information about three contacts. In these cases, in order to make inferences about the total number of contacts, we use within-respondent weights. For example, suppose respondent $i$ reports a total of $d_i = 6$ contacts, and provides detailed information about 3 of them. Then each of the three contacts receives a weight of $a_i = \frac{6}{3} = 2$. If, on the other hand, respondent $j$ reports a total of $d_j = 2$ contacts and provides detailed information about both of them, then $a_j = 1$. Conceptually, $a_i$ is the number of respondent $i$'s contacts represented by each

contact who gets reported about in detail[9], discusses this weighting approach in greater detail.

When we make population-level inferences about contact characteristics, such as the relationship and location distributions shown in Fig. 1, we use these contact weights in combination with the respondent weights[9]. For example, to estimate the proportion of contacts at work, we use

$$\widehat{p}_{\text{work}} = \frac{\sum_{i \in s} w_i\, a_i\, z_i^{\text{work}}}{\sum_{i \in s} w_i\, d_i},\tag{1}$$

where

- $s$ is the sample of all respondents
- $w_i$ is the respondent-level calibration weight
- $a_i$ is the within-respondent weight for respondent $i$'s contacts
- $z_i^{\text{work}}$ is a variable that has how many of respondent $i$'s detailed contacts were reported to have happened at work
- $d_i$ is the total number of contacts respondent $i$ reports

The intuition is that $w_i$ is the number of people respondent $i$ represents in the general population, and $a_i$ is the number of $i$'s contacts that is represented by each detailed contact.

**Statistical model.** To investigate factors associated with interpersonal contacts, we developed statistical models. We fit separate models to (1) the total reported contacts and (2) the number of reported non-household contacts. In each case, we model the expected number of contacts using a negative binomial distribution. The negative binomial distribution is appealing because it allows for overdispersion—that is, it enables us to model count data that exhibit more variance than would be expected under a Poisson distribution. This modeling approach has previously been used to study contact data[3].

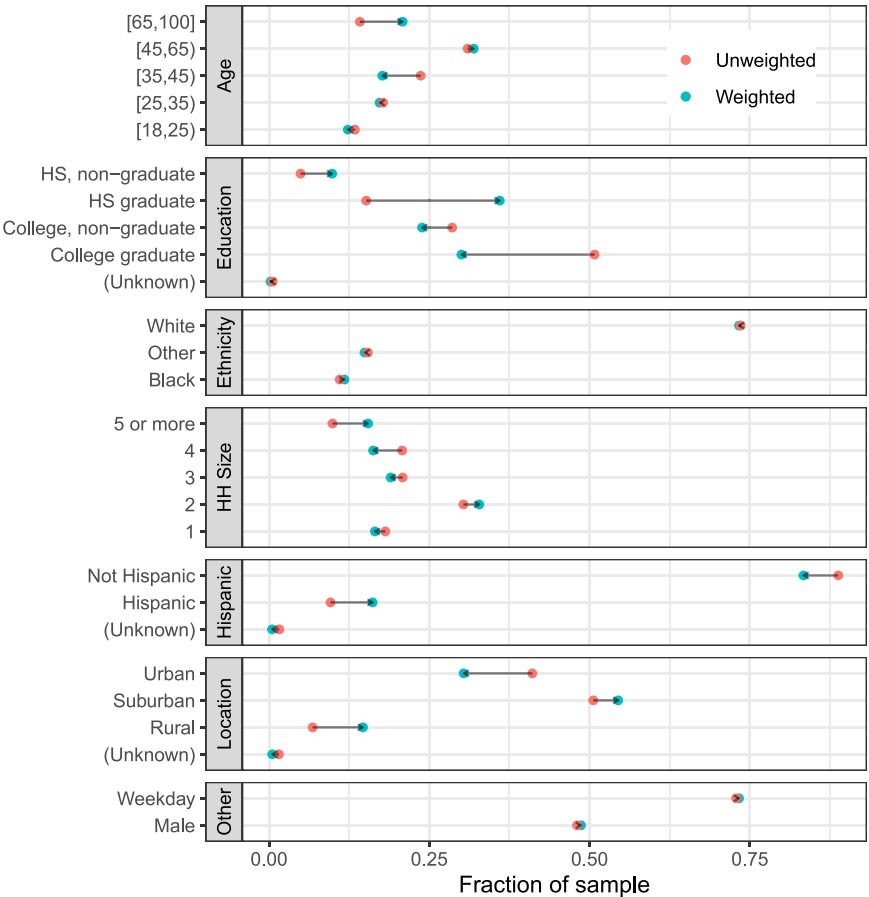

**Fig. 5 Characteristics of survey respondents.** We use calibration weights to improve the representativeness of our sample. Each facet shows the unweighted (red) and calibration weighted (blue) composition of survey respondents for a given covariate.

In our model, the log of the expected number of contacts for respondent $i$ is given by

$$\mu_i = \alpha + \mathbf{X}_i^{\mathrm{T}}\beta, \tag{2}$$

where $\mathbf{X}_i$ is a vector of covariates that includes age category, gender, household size, survey wave, city, race/ethnicity (Non-Hispanic White, Non-Hispanic Black/Hispanic/Non-Hispanic Other), and whether or not the day being reported about is a weekday. We include age by sex interactions, wave by race/ethnicity interactions, and wave by city interactions. $\beta$ is a vector of coefficients to be estimated.

Given $\mu_i$, we define $\lambda_i = \exp(\mu_i)$ to be the expected number of contacts for respondent $i$. Then we model the reported number of contacts for respondent $i$, $y_i$, as

$$y_i \sim \text{Neg-Bin}(\lambda_i, \phi), \tag{3}$$

where $\phi \in [1, \infty)$ is a shape parameter that is inversely related to overdispersion; that is, the higher $\phi$ is estimated to be, the more similar $y_i$'s distribution is to a Poisson distribution with rate parameter $\lambda_i$.

In our data, observations from Wave 0 are censored above 10, because the survey instrument allowed respondents to report up to "10 or more" contacts. Waves 1 and up allowed respondents to enter any number of contacts, but in this analysis we top-coded contacts at 29, following previous studies of contact data[3]. Reports that are topcoded or censored in any of the waves are treated as right-censored in the model. We adopt a Bayesian approach to fitting the model. For all of the regression coefficients $\beta$, we assume flat priors. For the intercept and the shape parameter, we assume very weak priors. Specifically, we assume a priori that the intercept $\alpha$ is distributed with mean 0 and a large variance by using $\text{pr}(\alpha) \sim$ Student-$t$ (3, 0, 10); and we assume a priori that the shape parameter $\phi$, is distributed with mean 1 and a very large variance by using $\text{pr}(\phi) \sim$ Gamma(0.01, 0.01). We did not collect data from Philadelphia in Wave 0, so the coefficient corresponding to Philadelphia in Wave 0 is constrained to be exactly 0 to allow estimation to proceed. Supplementary Table 1 shows summary statistics for the predictors used in our model.

Accounting for censoring, in our models the log posterior of the parameters given the data, $\log \text{pr}(\alpha, \beta, \phi | y, X)$, is proportional to

$$\log \text{pr}(\alpha, \beta, \phi | y, X) \propto \text{pr}(\alpha) + \text{pr}(\phi) + \Sigma_{i \in s_{nc}} w_i f_{\text{NB}}(y_i | \lambda_i, \phi) + \Sigma_{i \in s_c}[w_i(1 - F_{\text{NB}}(c_i | \lambda_i, \phi))] \tag{4}$$

where $s_{nc}$ is the set of responses that are not censored; $s_c$ is the set of responses that are right-censored, with response $i \in s_c$ being censored at value $c_i$; $f_{\text{NB}}(y | \mu, \phi) = \binom{y + \phi - 1}{y}\left(\frac{\mu}{\mu+\phi}\right)^y\left(\frac{\phi}{\mu+\phi}\right)^\phi$ is the PMF of the negative binomial distribution, and $F_{\text{NB}}$ is the cumulative distribution function $F_{\text{NB}}(y | \mu, \phi) = \sum_{x=0}^y f_{\text{NB}}(x | \mu, \phi)$; and $\lambda_i = \exp(\mu_i) = \exp(\alpha + \mathbf{X}_i^{\mathrm{T}}\beta)$ is the expected number of contacts or non-household contacts for respondent $i$. (The parameterizations of all distributions discussed here are the ones used in stan.) For each model, we run four chains of the sampler; each chain was run for 1000 warmup iterations and then 1000 sampling iterations. All $R$-hat statistics are 1, suggesting that the chains mixed effectively.

The model is nonlinear and has three sets of interacted predictors, making it challenging to directly interpret coefficient estimates. Therefore, in Fig. 2 and Supplementary Fig. 2 we show conditional effect plots and 95% credible intervals for covariates of interest. These plots illustrate model inferences by showing how the predicted number of contacts varies as a specific covariate varies. To do this, all other model predictors have to be held at fixed values. In Fig. 2 and Supplementary Fig. 2, we set predictors not being manipulated in each conditional effect plot to values for a white female aged 35–44 from the national sample with the average sample weight who lives in a two-person household during a weekday in wave 3. Supplementary Table 4 reports the actual coefficient estimates.

**Epidemiological model.** We estimate age-structured contact matrices for each wave of the BICS study. We group respondents and their contacts into six age bins: 0–18, 18–25, 25–35, 35–45, 45–65, and 65+. For each age group, we estimate the average daily number of contacts reported by respondents in that age group with contacts in every age group. In other words, our raw contact matrix, $\mathbf{M}$, has entries $m_{ij}$ which is the average number of daily contacts between respondents in age group, $j$, with their reported contacts in age group, $i$. Adjusting for survey weights,

we calculate $m_{ij}$ as

$$m_{ij} = \frac{\sum_{t=1}^{T_j} w_{t,j} y_{t,i}}{\sum_{t=1}^{T_j} w_{t,j}} \qquad (5)$$

where $w_{t,j}$ is the weight for reports made by participant $t$, in age group $j$, and $y_{t,i}$ is the number of reported contacts made by respondent $t$ in age group $i$. $T_j$ is the total number of respondents in age group $j$.

Contacts in the population must be reciprocal but due to differences in reporting in the survey our raw social contact matrix, $\mathbf{M}$, is not. We impose reciprocity by

$$c_{ij} = \frac{m_{ij} N_j + m_{ji} N_i}{2 N_j} \qquad (6)$$

where $c_{ij}$ are the entries of the reciprocal contact matrix, $\mathbf{C}$, and $N_i$ and $N_j$ the population size in age classs $i$ and $j$, respectively. For the youngest age group, for which we have no survey respondents, we assume

$$c_{i1} = \frac{m_{1i} N_i}{N_1}. \qquad (7)$$

These methods have been used previously to generate age-structured contact matrices from survey data[3–5,17,24].

We estimate within age group average number of contacts, $c_{ii}$, for the youngest age group by adapting methods from previous contact studies[5,17], and by using data from the United Kingdom POLYMOD study[3]. Specifically, for each wave of the BICS study, we calculate the ratio of the dominant eigenvalue for the contact matrix estimated from the BICS data to the dominant eigenvalue of the contact matrix from the POLYMOD study, with school contacts removed to reflect current school closures, for all age groups that are overlapping between the two studies. The within age group average number of contacts for the [0,18) group in the POLYMOD study is then scaled by this ratio to impute $c_{[0,18)[0,18)}$ in the BICS contact matrix.

The transmission dynamics of infectious diseases are summarized by the next-generation matrix, $\mathbf{N}$, that determines how an infection spreads when a pathogen is first introduced into a fully susceptible population. The basic reproduction number, $R_0$, is the average number of secondary infections arising from a single infection in a fully susceptible population, and is typically estimated as the spectral radius (dominant eigenvalue), $\rho(\mathbf{N})$ of the next-generation matrix, $\mathbf{N}$[25]. The $\mathbf{N}$ matrix is proportional to the population contact matrix, $\mathbf{C}$. The exact relationship between $\mathbf{N}$ and $\mathbf{C}$ is model-dependent, but for respiratory pathogens such as SARS-CoV-2, $\mathbf{N}$ is typically modeled as $\mathbf{C}$ scaled by the duration of infectiousness, $\frac{1}{\gamma}$, and the probability of transmission for a single contact, $q$. Therefore, the spectral radius of $\mathbf{N}$:

$$R_0 = \rho(\mathbf{N}) = \frac{q}{\gamma} \rho(\mathbf{C}) \qquad (8)$$

where $\rho(\mathbf{C})$ is the dominant eigenvalue of the reicprocal population contact matrix. In other words, $R_0$ is proportional to the dominant eigenvalue of $\mathbf{C}$.

Since $R_0$ is proportional to the dominant eigenvalue of $\mathbf{C}$, relative differences in $R_0$ under different contact patterns is equivalent to the ratios of the dominant eigenvalues of the different contact matrices. Specifically, if we assume that contact patterns in the population before physical distancing became widespread are equivalent to a baseline contact matrix, and that disease-specific parameters remained unchanged over the course of the survey period, the relative reduction in $R_0$ during physical distancing, compared to the baseline, is equivalent to the ratios of the dominant eigenvalues of the $\mathbf{C}$ matrices from the BICS study, $\mathbf{C}^{\mathrm{BICS}}$, to the dominant eigenvalue of the baseline pre-pandemic contact matrix $\mathbf{C}^{\mathrm{baseline}}$:

$$\frac{R_0^{\mathrm{BICS}}}{R_0^{\mathrm{baseline}}} = \frac{\rho(\mathbf{C}^{\mathrm{BICS}})}{\rho(\mathbf{C}^{\mathrm{baseline}})}. \qquad (9)$$

Further, if we assume a distribution for $R_0$ for COVID-19 in the absence of physical distancing, we can estimate the implied theoretical $R_0$ during the study period, by multiplying this ratio with the $R_0$ value in the absence of physical distancing. We assume that $R_0$ prior to physical distancing followed a normal distribution with mean 2.5 and standard deviation of 0.54 based on estimates from literature[5,26]. We vary the mean baseline $R_0$ value in sensitivity analyses. We compare the BICS contact matrices to two baseline business-as-usual scenarios: contact patterns estimated from a probability sample of US Facebook users[9] and contact patterns from the UK POLYMOD study[3], which has been widely used in many settings. We compute confidence intervals for the estimated $R_0$ by repeating the age-imputation and relative $R_0$ estimation on 5000 bootstrapped samples from the BICS, POLYMOD, and 2015 study contact matrices.

**Reporting summary**. Further information on research design is available in the Nature Research Reporting Summary linked to this article.

## Data availability
We have deposited our data in the Harvard Dataverse, https://doi.org/10.7910/DVN/M74AJ4[27].

## Code availability
Code to reproduce our analyses is available on GitHub at https://github.com/dfeehan/bics-paper-release[28].

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

## Acknowledgements
For helpful feedback on these results, we thank participants in the 1 April 2020 Berkeley Population Center Brown Bag, C. Jessica E. Metcalf, Caroline Buckee, and Audrey Dorélien. Seed funding was provided by a Berkeley Population Center pilot grant (NICHD P2CHD073964) and further funding was provided by the Hellman Fellows Program.

## Author contributions
D.M.F and A.S.M. designed the study, collected the data, conducted the analysis, and wrote the manuscript.

## Competing interests
The authors declare no competing interests.
