## [Peer Review File · Nature Communications]

REVIEWER COMMENTS

Reviewer #1 (Remarks to the Author):

This is a nicely written study with state of the art statistical analysis on the evolution and potential impact of contact patterns on Covid19 in the US.

While similar studies have been published recently in Europe (Latsuzbaia et al. & Jarvis et al.), this study is the first using data from the US and from a representative online panel.

I have the following comments:

- The authors need to discuss a major limitation of their study, whether mask wearing could have had an impact on results and at risk events of contact patterns. Are they planning to ask this question in future waves ?
- In addition to statistical model results, it would also be useful for the less statistically inclined reader to see "simple" average number of contacts in a table as a function of variables described in Table 1.
- The authors compare their results to a contact study conducted on Facebook in 2015 (e.g. in Figure 3). It would be great if this study was briefly explained in methods. The authors should also discuss if the nature of the survey tool (Facebook or vs. online panels) could have an impact on results.
- Abbreviations in figures should be explained in footnotes (e.g. non-hh in Figure 2, BICS in Figure 3, FB in Figure 4)
- The within respondent weight a_i is questionable as it is not obvious why the detailed contacts would be representative of the undetailed contacts. I suggest that they conduct a sensitivity analysis by limiting contacts to only detailed contacts with equal weight.
- The authors ought to compare and contrast their findings to those of Latsuzbaia et al. PLOS One (2020), which also conducted a similar online survey. E.g. They also found higher contact numbers in "minority" sections of the population.
- One of the strengths of the data is that the online panel is from all over the US, but this is not explored. Are there any regional differences in contact patterns as there seemed to be different waves of the epidemic in different states and different strategies for easing lockdown, potentially affecting contact patterns ?

Reviewer #2 (Remarks to the Author):

The general approach of this paper – using surveys to estimate changes in interpersonal contacts, then estimating the effects of those on epidemic dynamics – seems viable, but the approach taken is premised on a partially faulty premise that needs to be addressed.

Main Concern

Social contacts provide a necessary, but not sufficient condition for the potential of airborne disease transmission between interacting individuals. The insufficiency part is what's not adequately addressed in the current manuscript. In the context of an epidemic, removing social contacts does reduce the opportunities for transmission, because those contacts are necessary for exposure. So the part of the modeling exercise here that's looking at the effects of reductions in contacts on epidemic spread are likely appropriate. Unfortunately, the inverse does not necessarily hold – especially in a context where other behaviors are changing in such a way that the same

contacts no longer hold the same transmissibility potential. For example, in the current case of COVID-19, as people have increased their contacts towards a return to normal, they have also adopted other behaviors that substantially reduce the likelihood of transmission in any interpersonal contacts – e.g., mask wearing, increased handwashing, maintaining physical distance even when in social contact with others, testing and isolating such that the contacts had are selectively with people who are not infectious, etc. As a result, taking essentially a symmetrical approach to estimating the epidemic impacts of the observed social contact changes found in the survey is likely to overestimate the increases in R_0 that would correspond with those changes in contacts. Something needs to be done to address how much those other activities are likely to mediate the effects of increasing social contacts on estimated transmissions in the modeling strategy.

Lesser Concerns

There are a range of estimates of R_0 in the published literature, and the choice of 2.5 should be justified, and/or alternative scenarios modeled.

The claim on p. 4 that “There are surprisingly few existing estimates for the rate of contact in the US before the COVID-19 pandemic,” ignores the number of studies using the American Time Use Survey to provide such estimates.

Technically, you mean R_e or R_t in many of the places where you use R_0 .

Reviewer #3 (Remarks to the Author) :

The authors presented the results of an online survey carried out during the COVID-19 pandemic in the US, where they collected data on social contact patterns of individuals. The quantification of social contact patterns is critical to evaluate the effectiveness of the implemented non-pharmaceutical interventions, hence these data are definitely an important contributions, especially for researchers interested in modelling the transmission of SARS-CoV-2 in the US.

I have nonetheless a few comments that I hope can help the authors to improve the presentation and the discussion of their results of their survey.

Figure 1

1. In the caption of Figure 1, it is not clear to me the sentence “For example, the top panel shows that the average respondent reported almost 0.8 contacts to family members in Wave 2”. From the text, it seems that this sentence refers to panels A or B, when in fact the number 0.8 can be found in panel C. Could please the authors rephrase the caption to make it clearer?
2. I guess the vertical line in panels A and B is the median. Could this be written also in the caption?
3. Panel C (which is called in an ambiguous way “top panel” at the end of page 2) in Figure 1 shows the average number of contacts not at home with various types of contactees. I guess participants were asked, for each reported contactee, to explain the relationship to them, as well all the location where these persons were encountered. Now, among the different types of relationship to the participant, I notice also “spouse/romantic partner”, and “family”. I wonder if this means that these persons were only encountered outside of the home, or they were possibly met both at home and outside the home, but here only the extent to which they were met outside the home is reported. If these two groups of people are met also at home, I would say that, in epidemiological terms, it is not very relevant whether they are also met outside the home, as I imagine the contact at home being the longest in terms of duration. I would ask the authors to clarify this.
4. Panel D (called as “bottom panel” at the end of page 2) shows instead the visited locations outside the home. Hence, the category “Someone’s home” is the home of someone else, but for the participant is part of the general community, not his or her own home. Hence, I found it imprecise that the authors wrote at the beginning of page 3 that, across all waves, they found increases in the number of home contacts. According to the literature, starting from Mossong et al. (2008), contacts at someone’s home are not home contacts, rather contacts in other locations. I would thus rephrase the sentence, writing that increases in contacts at work and in the general community (or something else, but not at “home”) were found.

Figure 2

5. Would it be possible to add to each panels of Figure 2 the reference category? This would improve the clarity of the graph. If not possible, report at least the references for each variable in the caption. I find this particularly important for the variables age and city.

Results

6. I think what the authors are estimating in their epidemiological analysis is not the “basic reproduction number” R_0 , which is the number of secondary infections when one infective is introduced in a fully susceptible population, rather the “effective reproduction number” R_t , i.e., the average number of secondary cases generated by an infectious individual at time t (Liu et al. 2018 <https://www.pnas.org/content/115/50/12680>), which removes the assumption of a fully susceptible population.
7. I wonder how the effective reproduction number estimated by the authors is comparable to the time-dependent reproduction numbers estimated from case counts and their generation times (as used for monitoring the impact of interventions).
8. Another possible source of pre-pandemic contact data in the US are the matrices provided by Prem et al. (2017) (<https://journals.plos.org/ploscompbiol/article?id=10.1371/journal.pcbi.1005697#abstract0>), who build such matrices using available survey and demographic data for each country. Looking at the supplementary material, one can find also the social contact matrix for the US. Perhaps, a comparison with such matrix could be more appropriate than with the UK Polymod matrix. Or at least could be added for a third robustness check in Figure 4.
- 9.

Conclusions

10. During the US pandemic, there has been other efforts to collect contact data in the country, such as Dorélien et al. (2020), also cited by the authors, and Del Fava et al. (2020) (<https://www.medrxiv.org/content/10.1101/2020.05.15.20102657v1>). I would compare the results from the BICS survey to their results as well, in addition to those from China and the UK.

Methods

11. When discussing the epidemiological model in page 15, the authors report a formula to compute the average number of contacts between two age groups in the social contact matrix. It is not clear to me why the authors call y_i in the formula the weight of the reported contacts by participants in age group i . Should not the average m_{ij} be calculated as the weighted mean of the total number of contacts reported by respondents aged j with contactees aged i , namely, $y_{t,ji}$? In general, I find a good reference for the construction of social contact matrices to be Arregui et al. (2018) (<https://journals.plos.org/ploscompbiol/article?id=10.1371/journal.pcbi.1006638>).

Reviewer #1 (Remarks to the Author):

This is a nicely written study with state of the art statistical analysis on the evolution and potential impact of contact patterns on Covid19 in the US.

While similar studies have been published recently in Europe (Latsuzbaia et al. & Jarvis et al.), this study is the first using data from the US and from a representative online panel.

Thank you very much for your kind words, and for taking the time to review our paper. We appreciate your feedback.

I have the following comments:

- The authors need to discuss a major limitation of their study, whether mask wearing could have had an impact on results and at risk events of contact patterns. Are they planning to ask this question in future waves ?

Thanks to the reviewer for raising this important issue. For Waves 1-3 of our survey, we asked the respondent whether or not they were using a mask during a reported contact. There is considerable evidence to suggest that mask wearing reduces the risk of transmission and, as the reviewer notes, this will have an impact on the reproduction number. We have updated Figure 4 (also see below) to show the analysis with all contacts (as was originally shown) and with only contacts where no mask usage was reported. Note that these scenarios represent the two extreme ends of the spectrum of protection conferred by mask usage (from no reduction in transmission to complete reduction in transmission), and the actual reproduction number is likely somewhere in between.

Figure 4: R_0 estimates from the BICS contact matrices for each wave relative to two baseline contact matrices from the 2015 study and the UK POLYMOD study, and assuming a baseline R_0 value drawn from a normal distribution with mean 2.5 and standard deviation of 0.54. “FB” stands for the 2015 Facebook survey. Circles represent the reproduction number estimated with all reported contacts; diamonds represent the reproduction number estimated with only contacts where no mask usage was reported.

- In addition to statistical model results, it would also be useful for the less statistically inclined reader to see "simple" average number of contacts in a table as a function of variables described in Table 1.

This is a good point, thank you. As suggested, we have added Table 2, which has simple average numbers of contacts by the variables shown in Table 1.

- The authors compare their results to a contact study conducted on Facebook in 2015 (e.g. in Figure 3). It would be great if this study was briefly explained in methods. The authors should also discuss if the nature of the survey tool (Facebook or vs. online panels) could have an impact on results.

Good point, thank you. We have added a more detailed description of the 2015 survey, and we now make clear the additional assumption we need to compare results from the 2015 survey to

our sample; this can be found in the section “Comparison data from the 2015 survey,” which is in the Methods section.

- Abbreviations in figures should be explained in footnotes (e.g. non-hh in Figure 2, BICS in Figure 3, FB in Figure 4)

We have added definitions for several abbreviations at the end of the Figure captions. (We prefer this to footnotes, but would be happy to change this to match whatever the journal style dictates.) Specifically, we have added

- The definition of “Avg.” in Figure 1
- The definition of “BICS” in Figure 3
- The definition of “FB” in Figure 4

If the reviewer or editor would like us to add definitions for abbreviations elsewhere, we would be happy to oblige.

- The within respondent weight a_i is questionable as it is not obvious why the detailed contacts would be representative of the undetailed contacts. I suggest that they conduct a sensitivity analysis by limiting contacts to only detailed contacts with equal weight.

As the reviewer notes, our estimates for the location and relationship of respondents' contacts (shown in Figure 1c and 1d) make the assumption that the detailed contacts are a random sample of all of the respondents' contacts. To assess how sensitive our results are to this assumption, we compared the fraction of unweighted respondents who were

in each relationship and location for (i) all respondents (x axis); and (ii) only respondents who reported 3 or fewer contacts (so no a_i weights were needed -- y axis). The plots below show the results:

These plots make it clear that the reports that use weights and the reports that do not use weights are quite similar to one another: the points lie very close to the $y=x$ line. We therefore think this assumption is unlikely to have a big effect on our estimates. But we agree that this is worth being careful about, and we thank the reviewer for pushing us on this point.

- The authors ought to compare and contrast their findings to those of Latsuzbaia et al. PLOS One (2020), which also conducted a similar online survey. E.g. They also found higher contact numbers in "minority" sections of the population.

Thanks to the reviewer for this suggestion, and for pointing us to the study by Latsuzbaia et al. (2020) which we now cite in the paper and compare to our results. As the reviewer notes, Latsuzbaia et al. (2020) conducted a similar contact study in Luxembourg; they

found a 82% decline in average daily contacts compared to pre-pandemic levels. Other studies have also reported similar levels of decline in average daily contact. For example, Del Fava et al. (2020) find reductions in average daily contacts ranging from 48% in Germany to 85% in Italy, with Belgium, France, the Netherlands, and the UK having reductions between 73% and 75%. We have now added a discussion of these additional studies in our paper, in addition to the results from Jarvis et al. (74% decline for UK) and Zhang et al. (86% decline for Wuhan and 88% decline for Shanghai). The 82% decline in contacts in Wave 0 of our study is similar to reductions observed in other parts of the world under physical distancing.

- One of the strengths of the data is that the online panel is from all over the US, but this is not explored. Are there any regional differences in contact patterns as there seemed to be different waves of the epidemic in different states and different strategies for easing lockdown, potentially affecting contact patterns ?

The reviewer is correct that our national sample has respondents from across the US and, additionally, we oversampled in certain cities (see Figure 6). Exploring geographic patterns is an excellent suggestion.

To enrich the analysis in this paper, we expanded our statistical models to estimate more detailed city-specific information. Previously, we had included the city of the respondent as a covariate in our statistical model. To provide a deeper analysis of city-specific patterns, in this revision we interact predictors for each city with predictors for survey wave, allowing us to estimate changes in contact rates within each city over time. As the new version of Figure 2 shows, we see increases in contact rates in all cities, but with interesting differences in time patterns of increase in each one.

The reviewer's suggestion also touches on topics we are currently exploring in depth for future work. In ongoing work, we are (1) using the city specific age-structured contact matrices to simulate outbreak dynamics in each city to understand how local variations in contact patterns shape the transmission dynamics and mortality burden; and (2) developing methods for estimating state-specific contact matrices from our national sample. We agree with the reviewer that understanding spatio-temporal variations in contact patterns is important, but believe that doing so thoroughly is beyond the scope of this paper. We now mention this as an important avenue for future research.

Reviewer #2 (Remarks to the Author):

The general approach of this paper – using surveys to estimate changes in interpersonal contacts, then estimating the effects of those on epidemic dynamics – seems viable, but the approach taken is premised on a partially faulty premise that needs to be addressed.

Thank you very much for taking the time to review our paper. Based on your suggestions and feedback we have made a number of improvements to the manuscript.

Main Concern

Social contacts provide a necessary, but not sufficient condition for the potential of airborne disease transmission between interacting individuals. The insufficiency part is what's not adequately addressed in the current manuscript. In the context of an epidemic, removing social contacts does reduce the opportunities for transmission, because those contacts are necessary for exposure. So the part of the modeling exercise here that's looking at the effects of reductions in contacts on epidemic spread are likely appropriate. Unfortunately, the inverse does not necessarily hold – especially in a context where other behaviors are changing in such a way that the same contacts no longer hold the same transmissibility potential. For example, in the current case of COVID-19, as people have increased their contacts towards a return to normal, they have also adopted other behaviors that substantially reduce the likelihood of transmission in any interpersonal contacts – e.g., mask wearing, increased handwashing, maintaining physical distance even when in social contact with others, testing and isolating such that the contacts had are selectively with people who are not infectious, etc. As a result, taking essentially a symmetrical approach to estimating the epidemic impacts of the observed social contact changes found in the survey is likely to overestimate the increases in R_0 that would correspond with those changes in contacts. Something needs to be done to address how much those other activities are likely to mediate the effects of increasing social contacts on estimated transmissions in the modeling strategy.

This is an excellent point, similar to the one raised by Reviewer #1 above. To briefly recap our response: our survey actually asked about mask use for contacts during Waves 1 and up. Thus, in response to this feedback, we have added an analysis that estimates what R_0 would be if these masks were perfectly effective. In the updated version of our paper, in Figure 4 we show the analysis with all contacts (as was originally shown) and with only contacts where no mask usage was reported. Note that these scenarios represent the two extreme ends of the spectrum of protection conferred by mask usage (from no reduction in transmission to complete reduction in transmission), and the actual reproduction number is likely somewhere in between. Even when we restrict our analysis to only contacts where no mask usage was reported, we still see a general upward trend in the estimated reproduction number over time. We thank the reviewer for pushing us on this important point.

Lesser Concerns

There are a range of estimates of R_0 in the published literature, and the choice of 2.5 should be justified, and/or alternative scenarios modeled.

Thanks to the reviewer for this suggestion. Our choice of the distribution of R_0 was informed by estimates from Anderson et al. (2020) and a meta-analysis of the literature by Jarvis et al. (2020). As the reviewer correctly notes, there is a wide range of R_0 estimates in the published literature. We have now added a sensitivity analysis (Figure 11 and below), and repeated the analysis for estimating the reproduction number under physical distancing with higher and lower estimates of baseline R_0 values. We used the range of estimates of state-level R_0 (in the absence of physical distancing) reported in Pitzer et al. (2020). We find that for the highest baseline R_0 estimated by Pitzer et al. (2020) (5.17 for Missouri), the reduction in contacts is not sufficient to reduce R_0 to below 1 in Wave 0; otherwise the results remain qualitatively similar.

Figure 11: R_0 estimates from the BICS contact matrices for each wave relative to two baseline R_0 values and contact matrices from the 2015 study. The high baseline R_0 value is drawn from a normal distribution with mean 5.17 and standard deviation of 0.54. The low baseline R_0 value is drawn from a normal distribution with mean 1.92 and standard deviation of 0.54.

The claim on p. 4 that “There are surprisingly few existing estimates for the rate of contact in the US before the COVID-19 pandemic,” ignores the number of studies using the American Time Use Survey to provide such estimates.

This is a good point. Although estimates from time-use data are not directly comparable to our estimates (because, for example, time use surveys do not typically collect information about the ages of non-household contacts), they are still useful for understanding interaction in the context of disease transmission. Thus, we now cite Zagheni et al (2008) and a working paper by Dorelien et al (2020) which use time-use surveys to estimate contact patterns in the US. We agree that these papers would be interesting to readers of our study, so we thank the reviewer for pointing this out.

Technically, you mean R_e or R_t in many of the places where you use R_0 .

Thanks to the reviewer for raising this issue. In our study, we are relying on the age-structured contact matrices to estimate relative changes in the basic reproduction number (R_0) implied by the changing contact patterns over time. The “social contact hypothesis” assumes that the age-distribution of contacts is proportional to the age-distribution of exposed individuals for respiratory pathogens, such as SARS-CoV-2, that are spread through close interpersonal contact (Wallinga, Teunis, and Kretzschmar 2006). Under this assumption, relative changes in R_0 can be estimated by comparing the dominant eigenvalues of age-structured contact matrices as they are assumed to be proportional to the next-generation matrix (Wallinga, Teunis, and Kretzschmar 2006; Mossong et al. 2008; Klepac, Kissler, and Gog 2018; Jarvis et al. 2020). Thus, our estimated R_0 for each time point indicates the transmission potential for the pathogen in a fully susceptible population. Unfortunately, we cannot directly estimate the time-varying effective reproductive number (R_t) that measures the average number of secondary infections per case at each time-point in the epidemic, relying solely on contact structure and without information on the fraction of population that is susceptible. This is, however, an important distinction, and we have added some clarification in the paper.

Reviewer #3

The authors presented the results of an online survey carried out during the COVID-19 pandemic in the US, where they collected data on social contact patterns of individuals. The quantification of social contact patterns is critical to evaluate the effectiveness of the implemented non-pharmaceutical interventions, hence these data are definitely an important contributions, especially for researchers interested in modelling the transmission of SARS-CoV-2 in the US.

We thank the reviewer for this overview, and for the time taken to review our paper.

I have nonetheless a few comments that I hope can help the authors to improve the presentation and the discussion of their results of their survey.

Figure 1

1. In the caption of Figure 1, it is not clear to me the sentence "For example, the top panel shows that the average respondent reported almost 0.8 contacts to family members in Wave 2". From the text, it seems that this sentence refers to panels A or B, when in fact the number 0.8 can be found in panel C. Could please the authors rephrase the caption to make it clearer?

The reviewer is absolutely correct -- this was a typo: the sentence in question was in the wrong place. We have now fixed it; thank you for pointing this out.

2. I guess the vertical line in panels A and B is the median. Could this be written also in the caption?

Yes, the reviewer is correct - we agree that it would be clearer to add this to the caption, and we have now done so.

3. Panel C (which is called in an ambiguous way "toppanel" at the end of page 2) in Figure 1 shows the average number of contacts not at home with various types of contactees. I guess participants were asked, for each reported contactee, to explain the relationship to them, as well all the location where these persons were encountered. Now, among the different types of relationship to the participant, I notice also "spouse/romantic partner", and "family". I wonder if this means that these persons were only encountered outside of the home, or they were possibly met both at home and outside the home, but here only the extent to which they were met outside the home is reported. If these two groups of people are met also at home, I would say that, in epidemiological terms, it is not very relevant whether they are also met outside the home, as I imagine the contact at home being the longest in terms of duration. I would ask the authors to clarify this.

We agree that it is clearer to refer to Panel C and Panel D (as opposed to "top panel" and "bottom panel") -- we have made this change. Thank you.

The reviewer's comment has led us to add additional text to emphasize that the relationships and locations in Panels C and D refer to non-household contacts -- that is, to contact with people who are not living in the same household as the respondent. (This point is also relevant for the next comment, number 4). In the specific example the reviewer asks about -- "spouse/romantic partner" and "family" -- Panels C and D is showing the average number of contacts who fall in these two categories and who do not live in the same household as the respondent.

To try to make this clearer, we have added "non-household contacts" to the wording in the caption describing panels C and D; we had previously only mentioned this in the main text.

4. Panel D (called as "bottom panel" at the end of page 2) shows instead the visited locations outside the home. Hence, the category "Someone's home" is the home of someone else, but for the participant is part of the general community, not his or her own home. Hence, I found it imprecise that the authors wrote at the beginning of page 3 that, across all waves, they found increases in the number of home contacts. According to the literature, starting from Mossong et al. (2008), contacts at someone's home are not home contacts, rather contacts in other locations. I would thus rephrase the sentence, writing that increases in contacts at work and in the general community (or something else, but not at "home") were found.

To clarify, Panel D shows the locations of contacts that took place with people who do not live in the respondent's household. It is therefore possible that some of these 'home' contacts were in the respondent's own home, while some were in the home of someone else. We have added a footnote to help clarify this issue; the footnote reads:

"These are contacts respondents reported with people who do not live in their household. It is therefore possible that some of these 'home' contacts took place in the respondent's household; this would happen if, for example, neighbors came over to visit. They could also have taken place in someone else's household as would happen if, for example, the respondent had visited a friend at the friend's house."

We have modified the discussion of home contacts to include the results of Wave 3. Given that these contacts could potentially include the respondents own home, we have not changed the rest of our description of home contacts over time -- but we are open to making further changes if the reviewer thinks this is still unclear. Thank you for the feedback.

Figure 2

5. Would it be possible to add to each panels of Figure 2 the reference category? This would improve the clarity of the graph. If not possible, report at least the references for each variable in the caption. I find this particularly important for the variables age and city.

Good point, thank you. This comment, along with our expanded model, led us to change the way we illustrate model estimates. We now use conditional effect plots (Figures 2 and 7). Conditional effects plots allow us to show all categories for all of the predictors, and they also enable us to more easily show results from interacted predictors.

Although we think conditional effects plots most effectively summarize model inferences, some readers may still be interested in the actual coefficient estimates. Thus, we have also added a table that reports coefficients from each model (Table 3). In this table, we have added reference categories for all predictors, to help make interpretation as easy as possible.

Results

6. I think what the authors are estimating in their epidemiological analysis is not the “basic reproduction number” R_0 , which is the number of secondary infections when one infective is introduced in a fully susceptible population, rather the “effective reproduction number” R_t , i.e., the average number of secondary cases generated by an infectious individual at time t (Liu et al. 2018 <https://www.pnas.org/content/115/50/12680>), which removes the assumption of a fully susceptible population.

Thanks to the reviewer for raising this issue, which was also raised by Reviewer #2 (see above). To summarize our response above, in our study, we are relying on the age-structured contact matrices to estimate relative changes in the basic reproduction number (R_0) implied by the changing contact patterns over time under the “social contact hypothesis” assumption (Wallinga, Teunis, and Kretzschmar 2006; Mossong et al. 2008; Klepac, Kissler, and Gog 2018; Jarvis et al. 2020). Under this assumption, relative changes in R_0 can be estimated by comparing the dominant eigenvalues of age-structured contact matrices as they are assumed to be proportional to the next-generation matrix. Thus, our estimated R_0 for each time point indicates the transmission potential for the pathogen in a fully susceptible population. Unfortunately, we cannot directly estimate the time-varying effective reproductive number (R_t) that measures the average number of secondary infections per case at each time-point in the epidemic, relying solely on contact structure and without information on the fraction of population that is susceptible. This is, however, an important distinction, and we have added some clarification in the paper.

7. I wonder how the effective reproduction number estimated by the authors is comparable to the time-dependent reproduction numbers estimated from case counts and their generation times (as used for monitoring the impact of interventions).

Thanks to the reviewer for this suggestion, which is related to the previous question. As the reviewer notes in the previous question, the time-varying effective reproductive number (R_t) measures the average number of secondary infections per case at each

time-point in the epidemic (and therefore, not in a fully susceptible population). In our method, we estimate relative changes in the reproduction number over time from the age-structured contact matrices under the assumptions of the “social contact hypothesis” (Wallinga, Teunis, and Kretzschmar 2006; Mossong et al. 2008; Klepac, Kissler, and Gog 2018; Jarvis et al. 2020). Our estimated R_0 values indicate the potential for the outbreak to be sustained in a fully susceptible population. As such, our estimates are not directly comparable to estimates of R_t without additional information on the fraction of the population that are still susceptible at each time point.

8. Another possible source of pre-pandemic contact data in the US are the matrices provided by Prem et al. (2017) (<https://journals.plos.org/ploscompbiol/article?id=10.1371/journal.pcbi.1005697#abstract0>), who build such matrices using available survey and demographic data for each country. Looking at the supplementary material, one can find also the social contact matrix for the US. Perhaps, a comparison with such matrix could be more appropriate that with the UK Polymod matrix. Or at least could be added for a third robustness check in Figure 4.

Thanks to the reviewer for this suggestion. We have now added a comparison of the contact matrix from Prem et al. (2017) with the baseline pre-pandemic contact matrix in our study (from Feehan and Cobb (2019)) as well as the UK POLYMOD study (Mossong et al. (2008)). Figure 12 (also see below) shows the three contact matrices, and the average number of contacts by age group. We see very similar patterns of assortativeness of contacts by age, although the absolute levels of contact differ slightly across the three studies. We use the Feehan and Cobb (2019) study as our baseline as it relies on empirical measurements of the US population, rather than relying on model-based estimates (such as in Prem et al.). Nonetheless, we believe this is a useful comparison of available pre-pandemic contact patterns, and have included this figure in the Appendix.

Figure 12: Comparison of age-structured contact matrices from (A) Prem, Cook, and Jit (2017), (B) Feehan and Cobb 2019, and (C) Mossong et al. (2008). (D) Shows the average number of reported contacts for each age-group in the three studies.

Conclusions

10. During the US pandemic, there has been other efforts to collect contact data in the country, such as Dorélien et al. (2020), also cited by the authors, and Del Fava et al. (2020) (<https://www.medrxiv.org/content/10.1101/2020.05.15.20102657v1>). I would compare the results from the BICS survey to their results as well, in addition to those from China and the UK.

Thanks to the reviewer for this great suggestion, and for pointing us to the study by Del Fava et al. (2020) which we now cite in the paper. Del Fava et al. do not directly compare contacts over the course of the pandemic to pre-pandemic levels for the U.S., but do so for the countries in the original POLYMOD study. They find reductions in average daily contacts ranging from 48% in Germany to 85% in Italy, with Belgium, France, the Netherlands, and the UK having reductions between 73% and 75%. Latzubia et al. (2020) conducted a similar contact study in Luxembourg and found a 82% decline in average daily contacts compared to pre-pandemic levels. We have now added a discussion of these studies in our paper, in addition to the results from Jarvis et al. (74% decline for UK) and Zhang et al. (86% decline for Wuhan and 88% decline for Shanghai). The 82% decline in contacts in Wave 0 of our study is similar to reductions observed in other parts of the world under physical distancing.

Methods

11. When discussing the epidemiological model in page 15, the authors report a formula to compute the average number of contacts between two age groups in the social contact matrix. It is not clear to me why the authors call y_i in the formula the weight of the reported contacts by participants in age group i . Should not the average m_{ij} be calculated as the weighted mean of the total number of contacts reported by respondents aged j with contactees aged i , namely, $y_{t,j}$? In general, I find a good reference for the construction of social contact matrices to be Arregui et al. (2018) (<https://journals.plos.org/ploscompbiol/article?id=10.1371/journal.pcbi.1006638>).

The reviewer is correct that the average m_{ij} should be (and is) calculated as the weighted mean of the total number of contacts reported by respondents aged j with contactees aged i . After re-reading this section, we agree that the previous wording here was confusing, and we have edited it to try to make it more clear. We now write that “ $w_{t,j}$ is the weight for reports made by participant t , in age group j , and $y_{t,i}$ is the number of reported contacts made by respondent t in age group i .” Thank you for helping us make this improvement.

We also agree that the Arregui et al paper is a useful reference on this topic, and we have added a citation to it in the paper.

REVIEWERS' COMMENTS

Reviewer #1 (Remarks to the Author):

All comments and suggestions have been adequately addressed.

Reviewer #2 (Remarks to the Author):

This revision was relatively responsive to previous comments. I still think that the symmetrical nature of contact reductions/increases could be considered more carefully, but the mask-wearing addition provides a reasonable first approximation of this difference. That said, I think that the labels in Figure 4 are flipped for non-mask vs. mask wearing contacts (diamonds vs. circles) As presented it appears that non-masked contacts reduce R_0 more than those with masks, which isn't plausible.

Reviewer #3 (Remarks to the Author) :

I want to thank the authors for their impressive work on the revised version of this manuscript to meet the reviewers' comments and requests. I think the manuscript has greatly improved in terms of clarity and presentation of the results. In particular, I appreciated how they addressed my comments in the text.

I still have, however, a few minor comments that I would like to present to the authors, also in relation to the questions raised by the other reviewers:

1. Both Reviewer #2 and myself raised concerns about the name of the transmissibility index used to assess the impact of the changes in social contacts. The authors call this index as R_0 , the basic reproduction number; Reviewer #2 said that this should be called either as R_t or R_e , while I suggested to call it R_t , which I ambiguously defined in my previous comment as "effective reproduction number". The authors, in their reply, correctly stated that the calculation of the effective reproduction number, which is R_e , requires the information on the fraction of susceptible individuals in the population, which is not available. Hence, I searched more information regarding the difference between these epidemiological terms, both looking at the literature and asking the scientific opinion of colleagues in infectious disease epidemiology. I found out that the literature presents three possible indices: (i) the basic reproduction number, R_0 , which represents the average number of secondary infections generated by a typical index case in a fully susceptible population and relies on the assumption of the total absence of immunity in the population and the absence of any behavioral change and interventions; (ii) the effective reproduction number, R_e , which represents the average number of secondary infections generated by a typical index case in a PARTIALLY IMMUNE population, and therefore correct the R_0 for the fraction of susceptible population, which changes over time; (iii) the instantaneous/net reproduction number, R , which represents the average number of secondary infections generated by a typical index case in a population (either partially or fully susceptible), taking into account the current interventions and the potential spontaneous behavioral change in response to the risk of infection (Liu et al., 2018; The Royal Society, 2020; Zhang et al., 2020). This index is also denoted as $R(t)$ or R_t , to indicate that its values changes over time. On the basis of these definitions, I think the transmission index the authors estimate, similarly to what was done by similar studies (Coletti et al., 2020; Del Fava et al., 2020; Jarvis et al., 2020; Quaife et al., 2020), is neither the R_0 , because it is assumed that non-physical interventions were in place and people changed their behavior accordingly, nor R_e , because of the lack of information on the fraction of susceptible population. Rather, this estimation of this index is more in line with the definition of the net reproduction number R (or $R(t)$ or R_t , if one wants to show the dependency on time), and I would therefore call in such a way.
2. Pag. 19: the authors changed the formula and now the notation is indeed clearer. However, there is a typo in the text detailing the notation: the parameter $y_{t,i}$ should be the number of contacts made by respondent t (and not j) in age group i .
3. Regarding Table 2, added in the Appendix following a request by Reviewer #1, reports the average number of contacts from the data, overall and not at home, either without or with post-stratification weights. I think the information reported here would be more valuable if the standard deviation or the 95% CI were reported next to the average. Perhaps, the authors could just report the mean numbers of contacts from the weighted sample (therefore removing those from the unweighted sample) with an uncertainty measure next to them.

References

- Coletti, P., Wambua, J., Gimma, A., Willem, L., Vercruyssen, S., Vanhoute, B., Jarvis, C. I., Zandvoort, K. van, Edmunds, J., Beutels, P., & Hens, N. (2020). CoMix: Comparing mixing patterns in the Belgian population during and after lockdown. *MedRxiv*, 2020.08.06.20169763. <https://doi.org/10.1101/2020.08.06.20169763>
- Del Fava, E., Cimentada, J., Perrotta, D., Grow, A., Rampazzo, F., Gil-Clavel, S., & Zagheni, E. (2020). The differential impact of physical distancing strategies on social contacts relevant for the spread of COVID-19. *MedRxiv*, 2020.05.15.20102657. <https://doi.org/10.1101/2020.05.15.20102657>
- Jarvis, C. I., Van Zandvoort, K., Gimma, A., Prem, K., CMMID COVID-19 working group, Klepac, P., Rubin, G. J., & Edmunds, J. W. (2020). Quantifying the impact of physical distance measures on the transmission of COVID-19 in the UK. *BMC Medicine*, 18(1), 124. <https://doi.org/10.1186/s12916-020-01597-8>
- Liu, Q.-H., Ajelli, M., Aleta, A., Merler, S., Moreno, Y., & Vespignani, A. (2018). Measurability of the epidemic reproduction number in data-driven contact networks. *Proceedings of the National Academy of Sciences*, 115(50), 12680–12685. <https://doi.org/10.1073/pnas.1811115115>
- Quaife, M., van Zandvoort, K., Gimma, A., Shah, K., McCreesh, N., Prem, K., Barasa, E., Mwanga, D., Kangwana, B., Pinchoff, J., Bosse, N. I., Medley, G., O'Reilly, K., Leclerc, Q. J., Jit, M., Lowe, R., Davies, N. G., Deol, A. K., Knight, G. M., ... CMMID COVID-19 Working Group. (2020). The impact of COVID-19 control measures on social contacts and transmission in Kenyan informal settlements. *BMC Medicine*, 18(1), 316. <https://doi.org/10.1186/s12916-020-01779-4>
- The Royal Society. (2020). *Reproduction number (R) and growth rate (r) of the COVID-19 epidemic in the UK* (p. 86). <https://royalsociety.org/news/2020/09/set-c-covid-r-rate/>
- Zhang, J., Litvinova, M., Wang, W., Wang, Y., Deng, X., Chen, X., Li, M., Zheng, W., Yi, L., Chen, X., Wu, Q., Liang, Y., Wang, X., Yang, J., Sun, K., Longini, I. M., Halloran, M. E., Wu, P., Cowling, B. J., ... Yu, H. (2020). Evolving epidemiology and transmission dynamics of coronavirus disease 2019 outside Hubei province, China: A descriptive and modelling study. *The Lancet Infectious Diseases*, 20(7), 793–802. [https://doi.org/10.1016/S1473-3099\(20\)30230-9](https://doi.org/10.1016/S1473-3099(20)30230-9)

REVIEWERS' COMMENTS

Reviewer #2

This revision was relatively responsive to previous comments. I still think that the symmetrical nature of contact reductions/increases could be considered more carefully, but the mask-wearing addition provides a reasonable first approximation of this difference. That said, I think that the labels in Figure 4 are flipped for non-mask vs. mask wearing contacts (diamonds vs. circles) As presented it appears that non-masked contacts reduce R_0 more than those with masks, which isn't plausible.

Thanks to the reviewer for this additional feedback. We agree that the labels in Figure 4 are confusing. The circles show our estimated reproduction number for each wave when we consider all contacts recorded in the survey at each wave. The diamond shows our estimated reproduction number when we exclude all contacts where mask usage was reported and only include non-masked reported contacts (i.e. assume that masks are effective in reducing transmission). When we restrict reported contacts to this smaller number (without masks) we get a lower estimated reproduction number as expected. We have now changed the label to make this more clear.

Reviewer #3

I want to thank the authors for their impressive work on the revised version of this manuscript to meet the reviewers' comments and requests. I think the manuscript has greatly improved in terms of clarity and presentation of the results. In particular, I appreciated how they addressed my comments in the text.

I still have, however, a few minor comments that I would like to present to the authors, also in relation to the questions raised by the other reviewers:

1. *Both Reviewer #2 and myself raised concerns about the name of the transmissibility index used to assess the impact of the changes in social contacts. The authors call this index as R_0 , the basic reproduction number; Reviewer #2 said that this should be called either as R_t or R_e , while I*

suggested to call it R_t , which I ambiguously defined in my previous comment as “effective reproduction number”. The authors, in their reply, correctly stated that the calculation of the effective reproduction number, which is R_e , requires the information on the fraction of susceptible individuals in the population, which is not available. Hence, I searched more information regarding the difference between these epidemiological terms, both looking at the literature and asking the scientific opinion of colleagues in infectious disease epidemiology. I found out that the literature presents three possible indices: (i) the basic reproduction number, R_0 , which represents the average number of secondary infections generated by a typical index case in a fully susceptible population and relies on the assumption of the total absence of immunity in the population and the absence of any behavioral change and interventions; (ii) the effective reproduction number, R_e , which represents the average number of secondary infections generated by a typical index case in a PARTIALLY IMMUNE population, and therefore correct the R_0 for the fraction of susceptible population, which changes over time; (iii) the instantaneous/net reproduction number, R , which represents the average number of secondary infections generated by a typical index case in a population (either partially or fully susceptible), taking into account the current interventions and the potential spontaneous behavioral change in response to the risk of infection (Liu et al., 2018; The Royal Society, 2020; Zhang et al., 2020). This index is also denoted as $R(t)$ or R_t , to indicate that its values changes over time. On the basis of these definitions, I think the transmission index the authors estimate, similarly to what was done by similar studies (Coletti et al., 2020; Del Fava et al., 2020; Jarvis et al., 2020; Quaipe et al., 2020), is neither the R_0 , because it is assumed that non-physical interventions were in place and people changed their behavior accordingly, nor R_e , because of the lack of information on the fraction of susceptible population. Rather, this estimation of this index is more in line with the definition of the net reproduction number R (or $R(t)$ or R_t , if one wants to show the dependency on time), and I would therefore call in such a way.

We thank the reviewer for taking the time to provide this detailed feedback. As the reviewer notes there are multiple related, but not identical, measures of transmission potential reported in the literature. In our study, following the convention of Jarvis et al. (2020), Del Fava et al. (2020) and others, we estimate the theoretical reproduction number assuming a given baseline R_0 value. This implied theoretical reproduction number at each time point does not represent the effective reproduction number or the time-varying reproduction number as both of these measures imply some knowledge of the fraction of the population that is susceptible (this could be anywhere from 0 to 1). Rather our theoretical reproduction number, implied by the changed contact matrix under physical distancing, represents the transmission potential of the pathogen in a fully susceptible population (i.e. the start of the outbreak) if the current contact matrix remained unchanged. Therefore, while it accounts for the change in contact rates compared to baseline, it assumes that these changed rates remain the same. For example, the implied reproduction number in Wave 0 is the theoretical reproduction number at the start of an outbreak in a fully susceptible population, where the contact rate and pattern is given by the observed contact matrix in Wave 0. Similarly, the implied reproduction number in Wave 1 represents a hypothetical outbreak starting in a fully susceptible population with the contact matrix observed in Wave 1. It is perhaps easiest to imagine a new outbreak for each time point, that plays out in a fully susceptible population subject to the observed contact matrix at that time point; our implied reproduction number represents the transmission potential at the start of these outbreaks rather than a time-varying reproduction number over the course of the outbreak.

To estimate the net reproduction number or the effective reproduction number, as the reviewer notes, we would need a dynamic transmission model that accounts for both the change in fraction

susceptible over the course of the outbreak as well as the changing contact patterns under physical distancing. In ongoing work, we are using an age-structured dynamical model to estimate the time-varying reproduction numbers over the course of the outbreak - this requires a mechanistic transmission model and careful calibration of the model simulations to observed incidence and/or mortality data. This is beyond the scope of this current paper. While our simple approach here does not allow us to directly estimate these time-varying metrics, we still believe that our estimated reproduction numbers are an important signal of the transmission potential of the pathogen in the population at each time point. To make this distinction clearer in this paper, we now refer to our estimated reproduction number as the theoretical reproduction number implied by the contact matrix.

Pag. 19: the authors changed the formula and now the notation is indeed clearer. However, there is a typo in the text detailing the notation: the parameter $y_{i,i}$ should be the number of contacts made by respondent i (and not j) in age group i .

The reviewer is correct -- we have fixed this typo. Thank you.

- 2. Regarding Table 2, added in the Appendix following a request by Reviewer #1, reports the average number of contacts from the data, overall and not at home, either without or with post-stratification weights. I think the information reported here would be more valuable if the standard deviation or the 95% CI were reported next to the average. Perhaps, the authors could just report the mean numbers of contacts from the weighted sample (therefore removing those from the unweighted sample) with an uncertainty measure next to them.*

Thank you for this suggestion. We have expanded the Table into two: one table that shows weighted average numbers of contacts, with confidence intervals; and one table that shows unweighted average numbers of contacts, with confidence intervals.